# Ventricular Fibrillation and Tachycardia Detection Using Features Derived from Topological Data Analysis

Azeddine Mjahad *, Jose V. Frances-Villora [ID], Manuel Bataller-Mompean [ID] and Alfredo Rosado-Muñoz [ID]

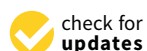



Processing and Digital Design Group, Department of Electronic Engineering, University of Valencia, 46100 Burjassot, Spain; jose.v.frances@uv.es (J.V.F.-V.); manuel.bataller@uv.es (M.B.-M.); alfredo.rosado@uv.es (A.R.-M.)
* Correspondence: mjahad.azeddine@uv.es

**Featured Application: Automated External Defibrillation (AED) and Implantable Cardioverter Defibrillators (ICD) require accurate algorithms to detect arrhythmias and discriminate among them. This work proposes specific features for algorithms implemented in such devices.**

**Abstract:** A rapid and accurate detection of ventricular arrhythmias is essential to take appropriate therapeutic actions when cardiac arrhythmias occur. Furthermore, the accurate discrimination between arrhythmias is also important, provided that the required shocking therapy would not be the same. In this work, the main novelty is the use of the mathematical method known as Topological Data Analysis (TDA) to generate new types of features which can contribute to the improvement of the detection and classification performance of cardiac arrhythmias such as Ventricular Fibrillation (VF) and Ventricular Tachycardia (VT). The electrocardiographic (ECG) signals used for this evaluation were obtained from the standard MIT-BIH and AHA databases. Two input data to the classify are evaluated: TDA features, and Persistence Diagram Image (PDI). Using the reduced TDA-obtained features, a high average accuracy near 99% was observed when discriminating four types of rhythms (98.68% to VF; 99.05% to VT; 98.76% to normal sinus; and 99.09% to Other rhythms) with specificity values higher than 97.16% in all cases. In addition, a higher accuracy of 99.51% was obtained when discriminating between shockable (VT/VF) and non-shockable rhythms (99.03% sensitivity and 99.67% specificity). These results show that the use of TDA-derived geometric features, combined in this case this the k-Nearest Neighbor (kNN) classifier, raises the classification performance above results in previous works. Considering that these results have been achieved without preselection of ECG episodes, it can be concluded that these features may be successfully introduced in Automated External Defibrillation (AED) and Implantable Cardioverter Defibrillation (ICD) therapies.

**Keywords:** electrocardiography analysis; ventricular arrhythmia detection; ventricular fibrillation detection; ventricular tachycardia detection; ECG signal classification; Topological Data Analysis; representation of point cloud; persistent diagram representation; landscape representation; silhouette representation

## 1. Introduction

A rapid and accurate detection of ventricular arrhythmias is essential to taking appropriate therapeutic actions. These pathologies are very common, being considered one of the main causes of death in developed countries, given that even weak episodes of Ventricular Fibrillation (VF) eventually cause sudden death.

Although arrhythmias have different origins, they can be considered a consequence of changes in cellular electrophysiology of the heart. Moreover, in most cases of sudden cardiac death, arrhythmogenic cardiac disorders appear as the main causes of death without showing evidence of pathological abnormalities of the heart.

To revert VF, the current protocol is the electrical defibrillation of the heart using an Automatic External Defibrillator (AED) [1], which can be commonly found nowadays in public places such as airports, shopping centers, sports arenas, etc. This process involves an external application of a high-energy electrical shock through the chest wall of the patient to allow the reinstatement of the normal rhythm. Some studies [2–4] have established that defibrillation success is conversely proportional to the time interval between the start of the Ventricular Fibrillation episode and the time when the electrical discharge is applied.

However, similar pathologies exist, like Ventricular Tachycardia (VT), requiring a different treatment than VF. In these cases, the signal may share some characteristics (lack of organization, irregularity, etc.) with VF, but the administration of an electrical shock to a patient not suffering VF could result in serious injuries or even bring about VF itself. This is why an accurate detection and classification of ventricular arrhythmias is so relevant.

The electrocardiogram (ECG) is an inexpensive and noninvasive tool used in the diagnosis of cardiac conduction disorders. It enables the analysis of the heart rate and morphology of different cardiac electrical waves, which, in turn, may permit the identification of various types of heart diseases. Because of this, ECG signals are considered an important and reliable source of information [5,6].

Many statistical methods have been applied to detect VF or VT using ECG data. However, following these manual methods, it is difficult to make a feature extraction capable of capturing the deep characteristics of ventricular arrhythmias. This is the reason why machine learning techniques have been effectively applied for the recognition of cardiac arrhythmias. In this sense, Orozco et al. [7] used the Wavelet method to detect ECG arrhythmias with three types of episodes (Normal, VT, and VF). In [8], Pooyan et al. used an SVM with Gaussian Kernel to detect ventricular abnormalities with morphological features. Tripathy et al. [9] detected and classified shockable (VF/VT) arrhythmias using Variational Mode Decomposition with Random Forest (RF) decision trees. In [10], Jekova et al. used fixed thresholds to implement a real-time detection of shockable episodes (VF/VT). In addition, in the same manner, other works harnessed other machine learning techniques for the detection and recognition of ventricular arrhythmias, as in Mohanty et al. [11], who used a C4.5 classifier; Jothiramalingam et al. [12], who employed a k-Nearest Neighbor (kNN) classifier; Tang et al. [13], who used Bayesian decision; or Kuzilez et al. [14], who employed Independent Component Analysis (ICA) and Decision Trees.

Over the last few years, there has been a general surge in the use of algebraic topology to analyze statistical data. Using this method, complicated data shapes can be categorized. Specifically, a commonly used topological method very used to extract features from a Point Cloud (a set of data points in space) is the Topological Data Analysis (TDA). TDA employs tools from algebraic and combinational topology to draw out properties that express data shapes. It can be considered a key method in attempting to interpret and comprehend characteristics that are otherwise unattainable through the use of other practices due to noise, dimension, or incompletion. It is so unique in its nature that TDA bridges the way between geometry and topology.

Successful and remarkable applications have been made in a varied selection of fields, and the range of applications continues to expand. Some of these applications include neuroscience [15], materials science [16], detection and quantification of periodic patterns in data [17,18], analysis of turbulent flows [19], natural language processing [20], or even detection and classification of breast cancers [21]. However, it has been used in image processing [22], computer vision [23], or signal and time series analysis [24,25].

Specifically, over the past few years, researchers have also begun to use TDA along with Machine Learning methods [26,27].

Within TDA, there is an important method called Persistence Homology that can be considered the main tool of TDA. As well as being a modification of the representation of homology using Point Cloud data, this method computes the homological characteristics of datasets.

In addition, TDA uses Persistence Diagrams and Persistence Barcodes to represent the abundant homological information about the shape of data. However, note that the use of algorithms of Machine Learning along with Persistence Diagrams or Barcodes is an area of TDA under research, looking for a way to alter these diagrams to be adaptable and congruous with Machine Learning methods. An alternative approach to these two diagrams is Persistence Landscapes.

In this work, we hypothesized that using Topological Data Analysis (TDA), some geometric features condensing relevant information about the 'shape of data' can be very valuable for the detection and discrimination of VF and VT rhythms, even in noisy and complex signals. Extracted features can then be applied to machine learning classifiers.

Thus, the goal of this work is to assess the improvement of the classification performance to detect and discriminate VF and VT episodes, when incorporating a set of TDA-derived geometric features in the feature extraction and selection stage. Note that the main difference with previous works is that these kinds of features have been never applied before in the analysis and classification of ventricular arrhythmias.

The main contributions of this work are

- The proposal of a novel classification procedure using features derived from Topological Data Analysis (TDA).
- The application of the proposed classification procedure to the detection and discrimination of VF and VT. Specifically, an accuracy near 99% is obtained.
- The application of the proposed classification procedure to the detection of shockable (VF/VT) and non-shockable rhythms. In this case, a 99.5% accuracy is obtained, the highest in the bibliography.
- The evidence that features derived from Topological Data Analysis can overcome conventional feature selection limitations by providing information about the 'shape of data' to the classifier.
- The high performance obtained without preselection of episodes shows that geometric features are good candidates to be incorporated into Automated External Defibrillator (AED) and Implantable Cardioverter Defibrillation (ICD) devices.

The paper is organized as follows. Section 2 is dedicated to the description of fundamental TDA. Section 3 introduces the dataset, explains the proposed methodology and details the used classification procedure. The results of the analysis and a discussion of these are presented in Sections 4 and 5, respectively. Finally, Section 6 concludes the paper.

## 2. Fundamental Concepts of TDA

This section outlines a simplified description of the mathematics behind Homology and Persistent Homology (PH). In TDA, cloud data are frequently seen as a simplicial complex, which is a set of points, lines, segments, triangles, and its n-dimensional counterparts. This allows one to use the methods from simplicial homology to quantify the shape of the data in terms of connections [28] and enables us to make a topological feature extraction. The process of topological feature extraction using PH can be summarized in the following steps:

- Data Point Cloud $\chi \in \mathbb{R}^n$ is employed as an input.
- For each data point (or vertex) $v_i \in \chi$, make $B(v_i)$ a ball of radius $\epsilon$ centered at each $v_i$, where $\epsilon \in \mathbb{R}^+$.
- Raise the value of $\epsilon$.
- A simplicial complex is built for each $\epsilon$ using Vietoris Rips and filtration.
- Measure PH and take note of its appearance and disappearance.
- Plot the ($\epsilon_{birth}$, $\epsilon_{death}$) appearance and disappearance coordinates for each PH on an extended real plane $\mathbb{R}^2 \bigcup \{\pm\infty\}$. The Persistence Diagram comes as an output.
- Lastly, the topological features are extracted.

In terms of mathematics, the input to a PH are the Point Cloud data. In the case of ECG, the input data are the time series. Taken's Delay Embedding Theorem can be used in

the conversion of time series data to point cloud data without losing topological properties. The approach consists of transforming a time series $x_t$, where $t \in \{1, 2, \ldots, T\}$, into its phase space representation. A point cloud or a set of points is obtained according to the following equation where $i = (1, 2, \ldots, T + n\tau)$ and $\tau$ is a delay parameter and $n$ specifies the dimension of the point cloud [29]:

$$v_i = x_i, x_{i+\tau}, \ldots, x_{i+n\tau} \tag{1}$$

Simplicial complexes are essential in the extraction of topological features from point cloud data. A single data point may define 0-Simplex. A line between two points denotes 1-Simplex. A triangle is a 2-Simplex. Tetrahedra represent 3-simplices (see Figure 1). Finally, a combination of simplices gives way to a Simplicial Complex called Vietoris Rips Complex [30–32].

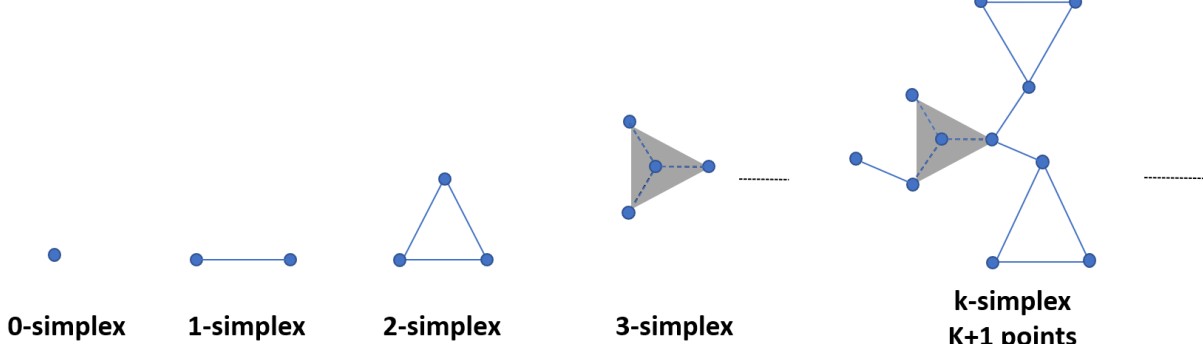

**Figure 1.** Gradual construction of various simplices (0-Simplex, 1-Simplex, 2-Simplex, ...) eventually gives way to a Simplicial Complex.

A simplicial complex can be taken from a dataset using the *Vietoris − Rips* construction. Being $X = (x_1, \ldots, x_n)$ a point cloud in an euclidean space $\mathbb{R}_n$, for each distance $\epsilon > 0$, represented by $VR(X; \epsilon)$, there is a simplicial complex with vertex set in X where $x_0, x_1, \ldots, x_k$ spreads a $k − simplex$ if the reciprocal distance between any pair of its varieties is smaller than $\epsilon$, where $d(x, x) \leq \epsilon$, for all $0 \leq i, j \leq k$.

When building a Simplicial Complex with Point Cloud data, it is needed to follow a set of rules. Firstly, a circle should be drawn with radius $\epsilon$ for each point in a point cloud. Then, when two circles intersect with each other and the radius is increased, a line is drawn to link the two points, which can be seen in Figure 2.

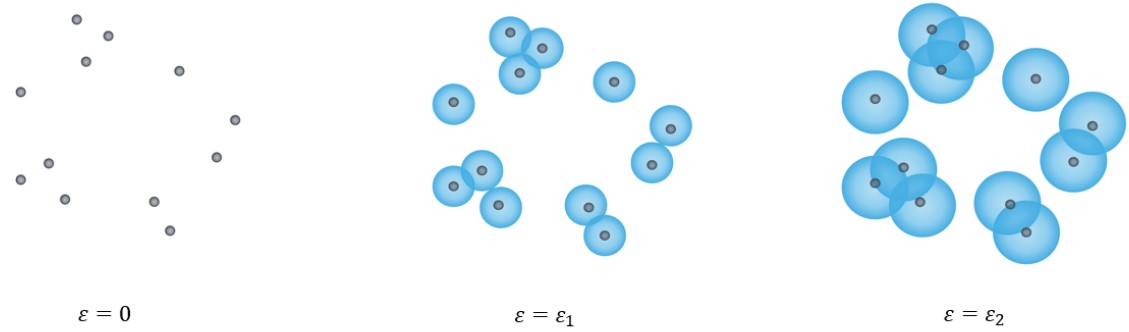

**Figure 2.** Diagram illustrating circular intersections and linked point clouds required to build a Simplicial Complex.

As $\epsilon$ gets longer, the Vietoris-Rips complex of a Point Cloud does, too. This is a filtration of simplicial complexes, i.e., a nested sequence of simplicial complexes, where

$VR(X; \epsilon)$, $\epsilon \geq 0$ satisfying $VR(X; \epsilon_1) \subseteq VR(X; \epsilon_2)$ if $\epsilon_1 \leq \epsilon_2$. To represent the distance between them, balls are drawn around each point. If two balls with radius $\epsilon$ intersect with each other, the two points are at a distance at most $2\epsilon$.

The Persistence Diagram representation (PDR) is a standard way to represent PH [33,34]. K-dimensional features consist of persistence diagrams; 0-dimensional features represent components that are connected, 1-dimensional features represent holes, 2-dimensional features voids, etc. [35]. Concurrently, a PDR $W_m$ is made of $n$ features, $W_{m_i} = (b_i, d_i)$, with $i = (1, 2, \ldots, n)$. Each point corresponds to the lifespan of one topological feature, where $b_i$ and $d_i$ are its birth time and death time, respectively (birth time indicates when the geometrical structure appears, while death time indicates when the geometrical structure disappears). Points are entirely located in the half-plane above the diagonal [36] (Figure 3).

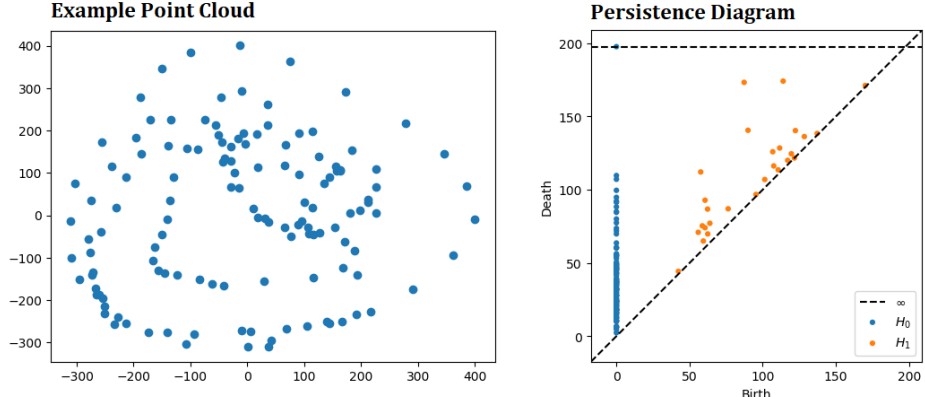

**Figure 3.** Representation of Point Cloud—RPC (**left**) on a Persistence Diagram representation—PDR (**right**).

When it comes to machine learning and statistics, a Persistence Landscapes Representation (LR) is more straightforward to work with than PDR and can be considered an alternative representation [37]. The approach takes the topological information that was previously encoded on a PDR and presents it as elements of a Hilbert space. Statistical learning methods can then be applied directly. Additionally, Persistence Silhouette representation (SR) [38] are constructed by mapping each point $z = (d, b)$ of a PDR to a piecewise linear function, namely the 'triangle' function $T_z$, which can be defined as follows:

$$T_z(y) = (y - b + d)l_{[b-d,b]}(y) + (b + d - y)l_{[b,b+d]}(y) \tag{2}$$

where $l_A(x)$ is the standard indicator function: $l_A(x) = 1$ if $x \in A$ and $l_A(x) = 0$, otherwise. A triangle function binds the points of the diagram to the diagonal, with segments parallel to the axes, and later they are rotated by 45 degrees. The triangles $T_z$ can be merged together in various manners, and if we take their $k_{Amax}$, i.e., the $k^{th}$ largest value in the set $T_z(y)$, the $k^{th}$ persistence landscape $\lambda^k = k - max_{zgD}T_z(y), k \in N^+$ results. The Persistence Landscape $\lambda_D$ is the gathering of functions $\lambda^k(y)$. Finally, the Power Weighted Silhouette representation $\Psi_p(t)$ (later named SR) is obtained by taking the weighted average of the functions $T_z(y)$, as the following equation shows.

$$SR = \Psi_p(t) = \frac{\sum_{zgD} \omega^p T_z(y)}{\sum_{zgD} \omega^p Z} \tag{3}$$

In Figure 4 we can see a representation of the PDR and the LR.

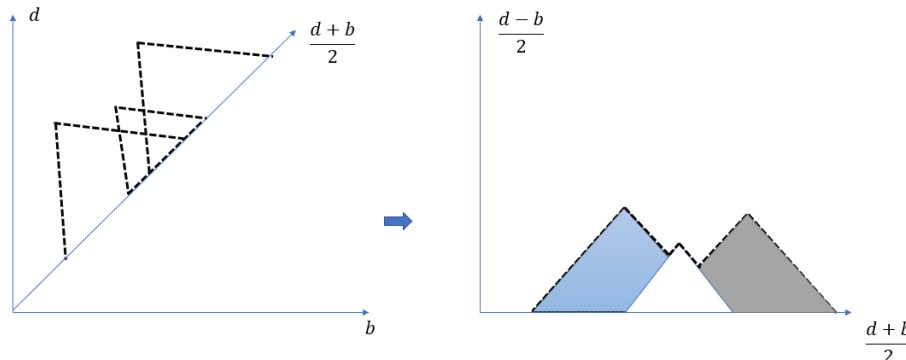

**Figure 4.** A visual example of the transformation of a persistence diagram representation (PDR) into a persistence landscape (PL). The horizontal axis represents birth time, while the vertical axis represents death time on the persistence diagram (**left**). The horizontal axis is the average of the homologies of the birth and death times, and the vertical axis is used for $(d - b)/2$ on the persistence landscape (**right**).

Another means of persistence diagram transformation is Persistence Images (PI) [23]. This allows for representations to be simply vectorized. Persistence images can be informally considered as a type of heatmap coming from a Calculate Gaussian KDE [39], which can be defined as follows:

$$\hat{f}(x) = \sum \alpha_i k(x - x_i) \tag{4}$$

where $k$ is kernel function centered at the data points $x_i$ with $i = (0, \dots, n)$, and $\alpha_i$ are the weighting coefficients.

## 3. Materials and Methods

To use the topological data features described above, the classification procedure proposed in Figure 5 is used.

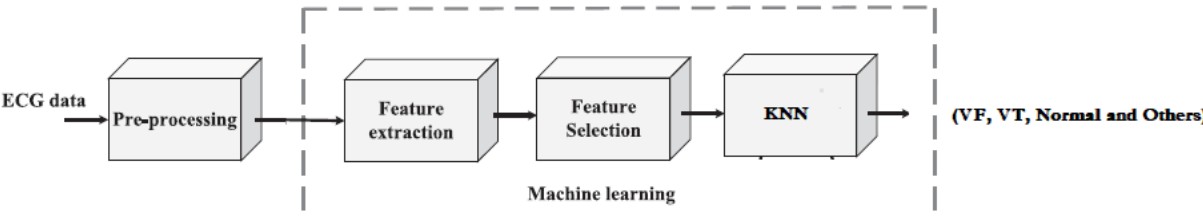

**Figure 5.** Schematic diagram illustrating the feature selection process proposed for the discrimination of ventricular arrhythmias, normal sinus, and other types of rhythms.

To provide a clear and detailed explanation of data processing, this section has been divided into different parts: Section 3.1 describes the used dataset; Section 3.2 details the noise cancellation, baseline removal, and segmentation is done as preprocessing; Section 3.3 describes the feature extraction and selection, and Section 3.4 outlines the classification procedure and parameters used to evaluate the performance of the classification.

### 3.1. Materials

Data records from two standard databases were used: MIT-BIH Malignant Arrhythmia Database [40,41] and AHA (American Heart Association) 2000 series [42]. It is important to note that no preselection of ECG episodes was done, i.e., all annotated segments from the database were used. Thus, 24 patients were analyzed (i.e., 24 records), 22 of which were from the MIT-BIH database and two additional patients from the AHA Database. Each record contained half an hour of continuous ECG recordings. According to the database

annotation, each segment was assigned to a class. Four classes of rhythms were established: Ventricular Fibrillation (named *VF*), including Ventricular Fibrillation or Ventricular Flutter episodes, Ventricular Tachycardia as *VT* class, sinus rhythms were assigned to the *Normal* class, and lastly, any signal not labeled within the above classes (e.g., other non-ventricular arrhythmias, noise, etc.) was assigned to class *Others*.

### 3.2. ECG Signal Preprocessing

The performance of machine learning algorithms can be brought down due to errors that may appear due to noise interruption or other input data corruption leading to improper feature values. Thus, a signal preprocessing is required to remove unwanted data corruption of the ECG signals: breathing, skin interference, baseline wander, powerline interference, motion artifact due to electrodes, muscle artifact, white Gaussian noise, etc. [43]. Since this work proposes the full data flow analysis from acquisition to classification as in a real scenario, we add this data preprocessing step, too. The steps used in the preprocessing stage to prepare signals for later processes are:

- Reduction of the baseline wandering, aiming to provide better quality and definition of the temporal signal, which will later result in better feature extraction. This stage involves the introduction of an 8th order infinite impulse response filter (IIR) with a Butterworth bandpass type ranging from 1 Hz to 45 Hz [44,45]. Figure 6 shows the effect of applying this bandpass filter, resulting in a reduction of the baseline.

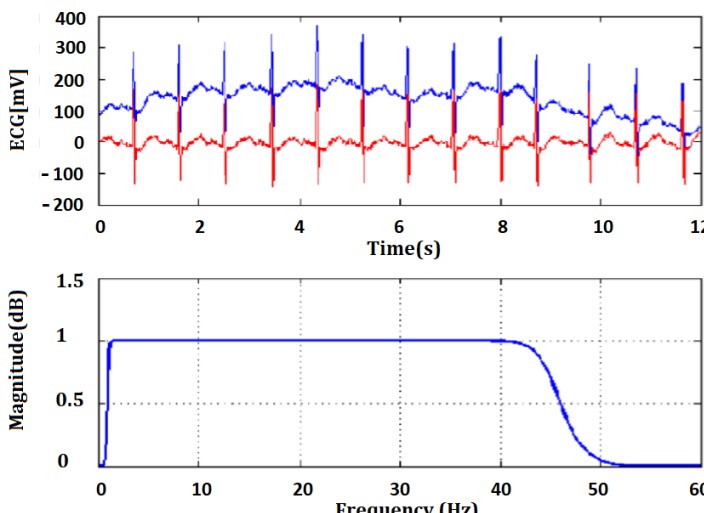

**Figure 6.** Bandpass filter application to a data segment from the *Normal* ECG class, and resulting baseline reduction.

- Later, a Window Reference Marks (WRM) and a time window (*tw*) are obtained. The mark indicates the start of each time window from which the features will be extracted. Consecutive time windows are obtained to analyze all ECG data. As the values between 50 and 120 beats per minute (bpm) can be considered a normal heart rate range [46], the minimum and maximum distance values between any two consecutive WRMs were established in 0.5 s and 1.2 s, respectively. Next, an algorithm already developed by the authors in [47] was used to obtain the calculation of WRM reference marks. A time window *tw* of 1.2 s (150 samples) in length was obtained, starting at each WRM reference mark, as the following equation shows, with $\{j = 1, 2, \ldots, NLMC\}$ where $NLMC$ is the number of local maxima LM marks existing in the ECG signal:

$$tw_i = [WRM_j, WRM_j + 1.2s] \tag{5}$$

- For each time window, the Taken's Delay Embedding Theorem is applied to convert the ECG data (a time series) to a Representation of Point Cloud data (RPC), a Persistence Diagram Representation (PDR), a Persistence Landscape (LR) and a weighted Silhouette Representation (SR).

### 3.3. Feature Extraction and Reduction

The feature extraction stage can be regarded as the most essential stage in the detection of ventricular arrhythmias. Within the methodology proposed, several discriminatory features from TDA were extracted. The temporal signal in each window was first transformed into Point Clouds using delay embedding. Then, topological representations were extracted: Persistence Diagram (PDR), Persistence Landscape (LR), and Power Weighted Silhouettes (SR) (Tables 1 and 2).

**Table 1.** Columns $a_1$, $a_2$, $a_3$ and $a_4$ correspond to the original ECG time signal windows; columns $b_1$, $b_2$, $b_3$, $b_4$ and $c_1$, $c_2$, $c_3$ and $c_4$ show RPC and PDR, respectively. Each row, from top to bottom, corresponds to *Normal*, *Other*, *VT* and *VF* classes, respectively.

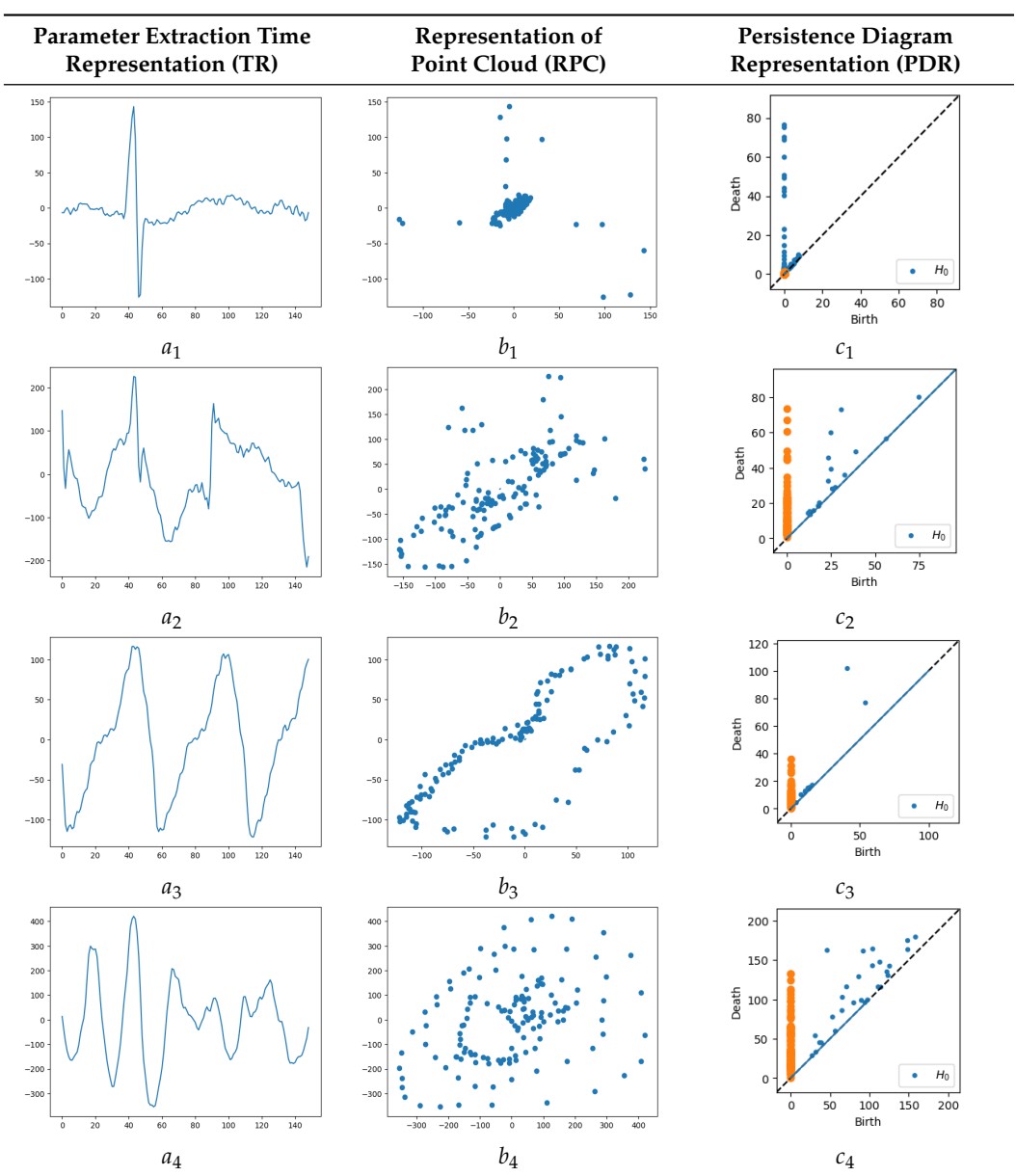

Table 1 illustrates the data point clouds (Representation of Point Cloud - RPC) and the persistence diagram (Persistence Diagram Representation - PDR) for each class (*Normal*, *Other*, *VT*, and *VF*). As seen, those representations provide a clear difference among classes. Regarding *Normal* class, the points in RPC have a focused distribution with respect to the rest of the arrhythmias where the points are scattered. Moreover, the point distribution differs between *VT* and *VF* as a very heterogeneous cloud is observed in *VF*, in contrast with *VT*. In the case of PDR, more points are located in a high birth-death ratio for *VF*, showing a clear difference with the rest of the rhythms. In Table 2, Persistence Diagrams (PDR) are compared with Persistence Landscapes Representation (LR) and Weighted Silhouettes Representation (SR). For each class, LR and SR show different shapes.

**Table 2.** From top to bottom, left to right: $a_1$, $a_2$, $a_3$ and $a_4$ illustrate Persistence Diagrams (PDR). $b_1$, $b_2$, $b_3$ and $b_4$ show Persistence Landscapes representation (LR), while $c_1$, $c_2$, $c_3$ and $c_4$ detail Weighted Silhouette representation (SR). These all correspond to the four classes *Normal*, *Other*, *VT* and *VF*, respectively.

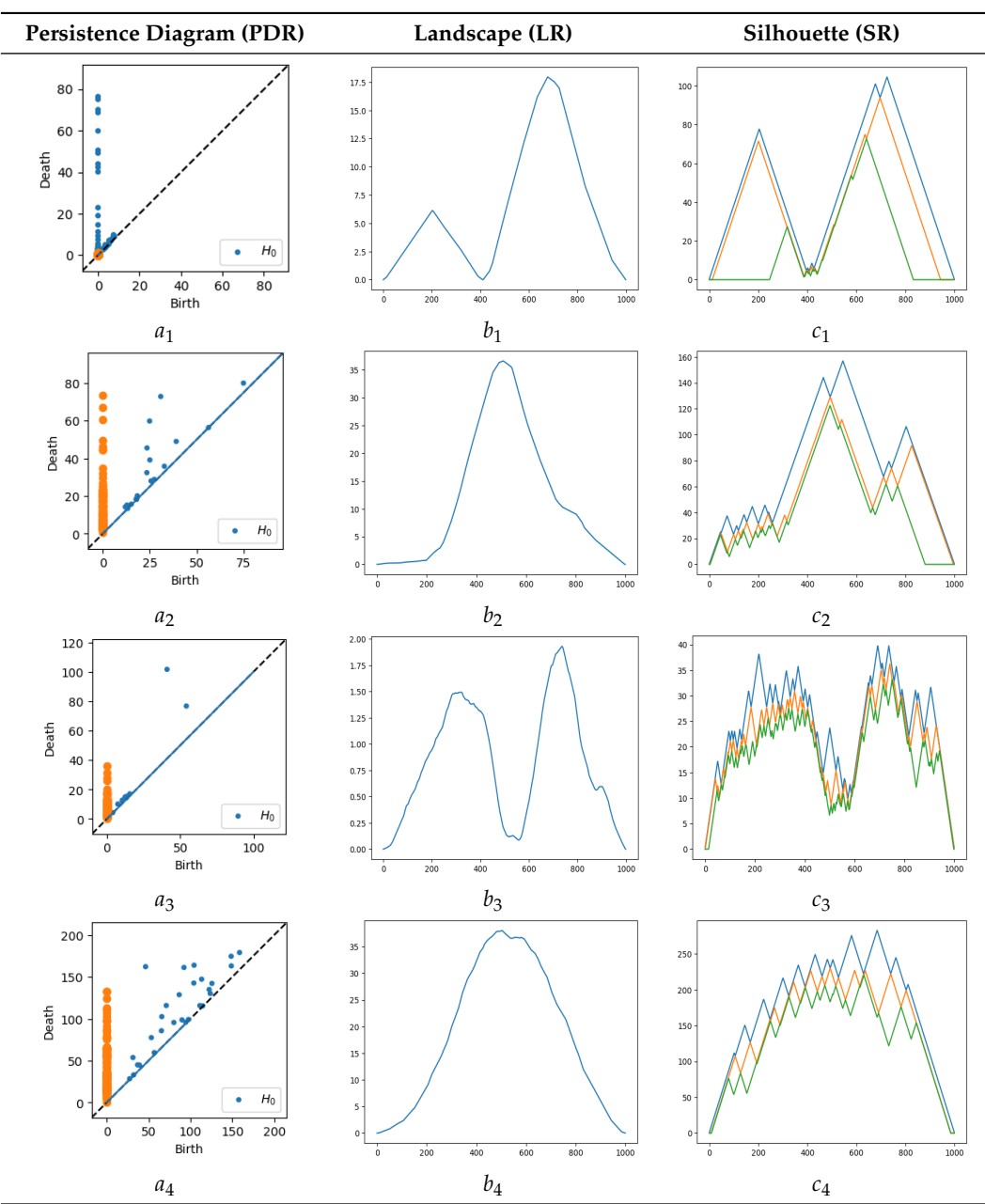

From each representation, a number of parameters are calculated, which will become the input features to the classifier. Initially, 79 parameters are evaluated, combining the most commonly used features in the bibliography with the proposed topological features. However, a feature reduction stage was performed. This stage allows the removal of any potentially redundant features existing, as well as reducing the computational complexity of the data analysis. In addition, we combined these features with other usual time-domain features representing statistical characteristics, such as variance, skewness, and kurtosis.

The feature selection was achieved using the Sequential Forward Selection (SFS) method, an iterative method that adds the best feature iteratively to the model until new additions do not improve the performance of the model. This method enables the selection of the most relevant features. Finally, a total of 27 features from all representations (time domain, RPC, PDR, LR, and SR) were selected amongst the 79 initial features. The extracted features are detailed in Table 3. This selection allows to improve the computational efficiency and reduce the generalization error of the model by removing irrelevant features or noise.

**Table 3.** List of extracted features using TDA and other time-domain parameters used as an input vector to the classifier.

| Representation or Domain | Parameter |
|---|---|
| Time Domain | Std (Standard deviation along the specified axis) [48] <br> Permutation entropy [49] <br> Spectral entropy [50] <br> Singular Value decomposition entropy [51] <br> Aproximate entropy [52] <br> Sample entropy [53] <br> Lempel-Ziv complexity [54] <br> Shannon entropy [55] <br> Petrosian fractal dimension [56] <br> Katz fractal dimension [57] <br> Higuchi fractal dimension [58] <br> Detrended fluctuation analysis [59] |
| Representation of Point Cloud (RPC) | Std (Standard deviation along the specified axis) [53] <br> Tsem (trimmed standar error of the mean) [60] <br> Nanmean [61] <br> Tvar (Tail value at risk) [62] |
| Persistence Diagram Representation (PDR) | Persistence Weighted Gaussian Kernel [63] <br> Approximate PWG kernel [63] <br> Persistence Scale SpaceKerne [64] <br> Approximate PSS kernel [64] <br> Sliced Wasserstien Distance [27] <br> Sliced Wasserstein Kernel [27] |
| Landscape Representation (LR) | Tsem (Trimmed standard error of the mean) [60] <br> Tstd (Trimmed sample standard deviation) [53] <br> Wasserstien distance [65] <br> Heat kernel distance bottleneck [53] |
| Silhouette Representation (SR) | RMS (Root Mean Square) [66] |

In addition, another input feature set was obtained. In this case, the Calculate Gaussian KDE was applied to the Persistence Diagram, and then an image-like was obtained (Table 3). The obtained image will be used as a direct input to the classifier, in a similar form as in [47]. Table 4 shows the resulting Gaussian KDE from a PDR.

**Table 4.** Persistence Diagrams (left) $a_1$, $a_2$, $a_3$, $a_4$ and representation of Calculate Gausssian KDE (right) $b_1$, $b_2$, $b_3$, $b_4$ for classes *Normal*, *Other*, *VT* and *VF*, respectively. The representation of KDE is used as input to the kNN classifier in the proposed PDI method.

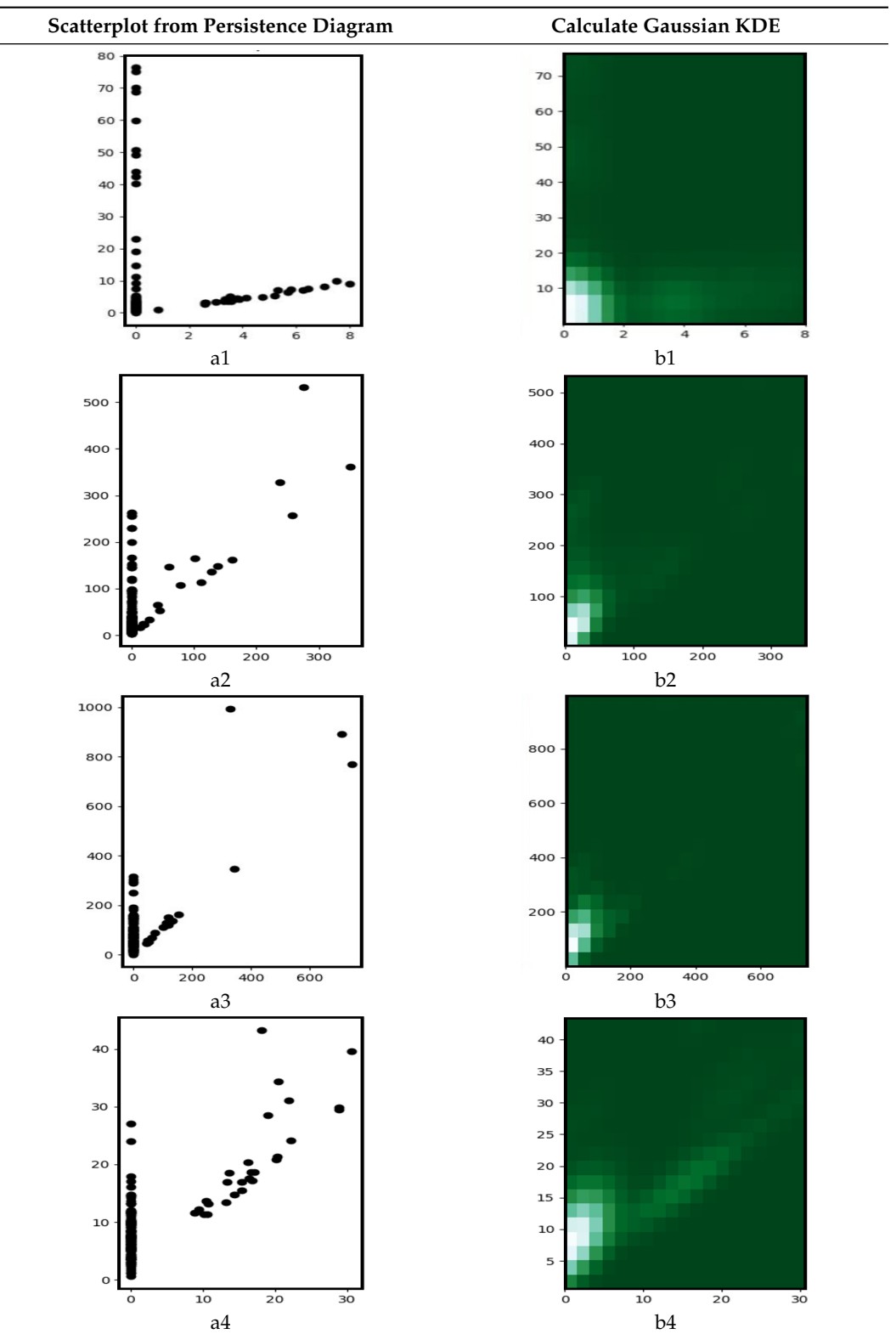

### 3.4. Classification Procedure and Performance Evaluation

This work used supervised learning. The input to the classifier was formed by a feature vector linking together all the selected features calculated from consecutive ECG

time windows. The only classifier used in this work was the k-Nearest Neighbor (kNN) algorithm, which is commonly used in the bibliography [12,67]. And the used distance is the Euclidean distance. In addition, two different input sets are evaluated. The first input set corresponds to the features described in Table 3, named the TDA method. The second input set is based on the Gaussian KDE bidimensional representation, named Persistence Diagram Image - PDI.

It has been used as a repeated random sub-sampling validation technique. Thus, for each class, 67% of data was randomly selected for training and the remaining 33% for testing. The kNN training process was done, and then the testing dataset was used to evaluate the classification performance by measuring the Sensibility (Sen), Specificity (Spe), and Accuracy (Acc). This cross-validation approach was repeated five times at random, and the performance of the classifiers was evaluated overall by taking this five iterations average. This number of iterations was chosen after some trials as it showed the lowest generalization error.

Standard statistical parameters were followed to assess the performance in accurate classification of the ECG signal into the *VF*, *VT*, *Normal*, or *Others* classes. These include the Sensitivity (Sen), Specificity (Spe), and Accuracy (Acc), which are calculated using the following equations where *TP*, *TN*, *FP*, and *FN* represent the number of true positives, true negatives, false positives, and false negatives, respectively:

$$Sensitivity = \frac{TP}{TP + FN} \times 100\% \qquad (6)$$

$$Specificity = \frac{TN}{TN + FP} \times 100\% \qquad (7)$$

$$Accuracy = \frac{TP + TN}{TP + FP + TN + FN} \times 100\% \qquad (8)$$

## 4. Results

The experiments were carried out using signals from the MIT-BIH and AHA standard databases, Section 3.1. They were divided into four classes, namely *VF*, *VT*, *Normal* and *Others*. The preprocessing stage carries out an 8th order bandpass (1 Hz to 45 Hz) Butterworth IIR filter to denoise and reduce the baseline variation, Section 3.2, and calculates the window reference marks (WRM) of the signal, marks indicating the beginning and end of the 1.2 s time window from the temporal signals.

At the feature extraction, we have proposed two different topological techniques to extract the parameters feeding the classifier: Topological Data Analysis (TDA) and Persistence Diagrams (PDI). In the case of the TDA method, each window of temporal signals were converted first into a Point Clouds representation, using delay embedding, and then into Persistence Diagrams, Persistence Landscapes, and Power Weighted Silhouettes. Finally, some parameters were extracted from these diagrams, Section 3.3, and then combined to create the features vector feeding the input of the classifier. Concerning the PDI method, the gaussian KDE was applied to the Persistence Diagram and the whole resulting image was used as a direct input to the classifier.

The k-Nearest Neighbor (kNN) classifier was the only classifier used for both proposals.

For each class, 67% of data was randomly selected for training and the remaining 33% for testing. The kNN training process was calculated and then the testing dataset was used to evaluate the classification performance by measuring the Sensibility (Se), Specificity (Sp), and Accuracy (Acc). This approach was repeated five times at random, and the performance of the classifiers was evaluated overall by taking this five iterations average. This number of iterations was chosen after some trials because it showed the lowest generalization error.

Table 5 shows the confusion matrix for one of these iterations. It shows a great classification performance. Nevertheless, the values represented in the following tables (Tables 6–9) indicate the average performance values obtained from the repeated random validation used in this work.

**Table 5.** Confusion matrix for classification of *VF*, *VT*, *Others*, and *Normal* classes using the TDA topological method.

| Algorithms | TDA | | | |
|---|---|---|---|---|
| | VF | VT | Normal | Other |
| VF | 1701 | 36 | 12 | 3 |
| VT | 45 | 601 | 9 | 1 |
| Normal | 15 | 3 | 4957 | 28 |
| Other | 0 | 2 | 55 | 1940 |

Thus, Tables 6–9 show the testing classification results for TDA and PDI feature selection methods. As it can be seen, the TDA method shows better classification results than the PDI. On the one hand, the PDI method results in values of accuracy above 92% for all classes, having better accuracy values for *VT* and *Other* classes (97.38% and 96.19%, respectively), but curiously falling to 92.65% for the detection of *Normal* sinus rhythms. The sensitivity widely varies depending on the case, ranging from 82.25% for *VT* to 93.09% for the *Normal* classes, being more sensitive to *Normal* and *Others* classes (around 93%) than to *VT* and *VF* (around 84%). Except for the *Normal* case, the global specificity (Spe) becomes greater than sensitivity, reaching the value of 98.53% for the *VT* class.

On the other hand, the TDA method results in very high results of accuracy, around 99% for all classes, with little differences between them. The sensitivity remains above 97% except for the *VT* class, falling to 92.72% and getting the maximum sensitivity value for the *Normal* case (99.05%), with 97.07% for the *VF* class. Finally, the global specificity achieves high values: near 99% for the *Normal* class and above 99% for the rest of the classes, hitting a maximum of 99.53% for the *VT* class.

**Table 6.** Results obtained for *VF* class classification in testing.

| Type | VF | | | | | |
|---|---|---|---|---|---|---|
| Algorithms | Sensitivity% | | Specificity% | | | Accuracy% |
| | VF | Global | VT | Other | N | Total |
| TDA | 97.07 | 99.25 | 93.78 | 99.90 | 99.68 | 98.68 |
| PDI | 84.34 | 96.77 | 89.70 | 99.14 | 96.68 | 94.26 |

**Table 7.** Results obtained for *VT* class classification in testing.

| Type | VT | | | | | |
|---|---|---|---|---|---|---|
| Algorithms | Sensitivity% | | Specificity% | | | Accuracy% |
| | VT | Global | VF | Other | Normal | Total |
| TDA | 92.72 | 99.53 | 97.93 | 99.89 | 99.94 | 99.05 |
| PDI | 82.25 | 98.53 | 94.85 | 99.62 | 99.36 | 97.38 |

**Table 8.** Results obtained for *Normal* class classification in testing.

| Type | Normal | | | | | |
|---|---|---|---|---|---|---|
| **Algorithms** | **Sensitivity%** | | **Specificity%** | | | **Accuracy%** |
| | **Normal** | **Global** | **VF** | **VT** | **Other** | **Total** |
| TDA | 99.05 | 98.45 | 99.27 | 98.88 | 97.16 | 98.76 |
| PDI | 93.09 | 92.14 | 90.22 | 91.65 | 93.95 | 92.65 |

**Table 9.** Results obtained for *Other* class classification in testing.

| Type | Other | | | | | |
|---|---|---|---|---|---|---|
| **Algorithms** | **Sensitivity%** | | **Specificity%** | | | **Accuracy%** |
| | **Other** | **Global** | **VT** | **Normal** | **VF** | **Total** |
| TDA | 97.43 | 99.54 | 99.88 | 99.40 | 99.82 | 99.09 |
| PDI | 92.86 | 97.15 | 99.02 | 96.76 | 97.72 | 96.19 |

## 5. Discussion

The same as with any other classification problem, the detection of ventricular arrhythmias normally uses a feature extraction and selection stage to optimize the class separation capabilities of the classifier. This feature selection stage aims at gathering the relevant aspects of the ECG signal based on TDA. Among a wide set of features, a reduction stage is done to lower the number of features used as input to the classifier.

In this work, we hypothesized that, by using Topological Data Analysis (TDA), some geometric features containing information about the 'shape of data' could be extracted. This method condenses the relevant information about the shape of the data, resulting in very valuable for the detection and discrimination between shockable VF and VT rhythms, even in noise and complex signals cases.

The obtained results (Tables 6–9) use the kNN classifier with the input features obtained by using two topological methods (TDA and PDI). Results show that the TDA features provide better results. For this reason, the TDA method is compared with other works in the bibliography. We have used the kNN classifier, given that it is enough to prove the improvement in classification results compared to other works. Nevertheless, using other classifiers is an open topic, which may lead to reach even better classification results.

As it can be seen from Tables 6–9, the use of the proposed TDA method provides an average accuracy of 98.9% for multiclass discrimination, which differentiates *VF* and *VT* ventricular arrhythmias but also *Normal* and *Other* types of rhythms. On the other hand, Table 10 shows a two-class classification approach to show that the proposed TDA method provides an accuracy of 99.5% when used to discriminate shockable (*VT* or *VF*) and non-shockable rhythms (rest of cardiac rhythms).

Thus, it can be established that the TDA method provides a very high classification performance. Nevertheless, we show a comparison of results with other works in the bibliography. Note, however, that this comparison is difficult due to the differences in the source signals used by different works; or even in the type of discrimination, they carry out: some works discriminate between ventricular arrhythmias and non-ventricular rhythms, others between ventricular fibrillation rhythms and non-ventricular fibrillation, others between shockable rhythms and non-shockable rhythms (considering as shockable both VT and VF).

For this reason, we divide the comparison into two separate blocks: the first block focuses on the comparison with works performing rhythm discrimination, while the second focus on the comparison with those works performing shockable vs. non-shockable signal discrimination.

Table 11 shows a group of works distinguishing between VF and non-VF rhythms. In this group, Roopaei et al. [68] obtained an accuracy of 88.60% using chaotic-based reconstructed phase space features to detect VF episodes. Arafat et al. [69] achieved a high value in the specificity of detecting VF episodes (Sp = 98.51%) using an improved version of the Threshold Crossing Interval (TCI) algorithm, called TCSC, and the MIT-BIH and CUDB databases. However, this detection was carried out with a sensitivity as low as 80.97%. Later, Alonso-Atienza et al. [70] obtained high values of specificity and accuracy (Spe = 97.10% and Acc = 96.80%) for the discrimination of VF episodes, with their specific feature selection and SVM classifiers. In their case, the sensitivity got a moderate value of 91.90%. Further, Li and Rajagopalan [71] used a genetic algorithm to make the feature selection for classifying VF episodes, achieving high-performance values: Sen = 98.40%, Spe = 98.00%, and Acc = 96.30%. Next, Acharya et al. [72] obtained high-performance values of specificity (Spe = 98.19%) and accuracy (Acc = 97.88%) using a Convolutional Neural Network (CNN) for the detection of VF. However, they achieved an extremely low value of sensitivity (Sen = 56.44%). Finally, in 2019 Ibtehaz et al. [73] got the highest results in this group, using a scheme of incorporating Empirical Mode Decomposition (EMD) and SVM classifiers (Sen = 99.99%, Spe = 98.40%, Acc = 99.19%) for the classification of VF and non-VF episodes.

As it can be seen, the results of the TDA proposal in this work achieve one of the best results (Sen = 97.07%, Spe = 99.25%, and Acc = 98.68%) compared with other works of the VF-discriminating group, with the only exception of Ibtehaz [73], that obtained better results. However, to establish a fair comparison, note that Ibtehaz obtained slightly higher results (i.e., a difference of 0.51% in Accuracy) at the expense of preselecting and rejecting the noise episodes, while in this work, there was not any preselection of ECG episodes.

Furthermore, another group of works in the bibliography can be compared, distinguishing between VT and VF rhythms (Table 11). In this group, Xie et al. [74] proposed a fuzzy similarity-based approximate entropy approach, distinguishing between VT and VF and obtaining high-performance ratios (Sen = 97.98% and Spe = 97.03% to VF and Sen = 97.03% and Spe = 97.98% to VT). However, to establish a fair comparison, it must be considered that Xie was selected as input data representative and clean episodes of VF and VT, while our work was done without preselection of ECG episodes. This kind of preselection is usual in the literature, as in Kaur and Singh [75], that used a selection of VF and VT episodes from the MIT-BIH database, using Empirical Mode Decomposition (EMD) and Approximate Entropy. Kaur and Singh obtaining moderate values for classification performance (Sen = 90.47%, Spe= 91.66%, and Acc = 91.17%). Later, Xia et al. [76] obtained high performance values (Sen = 98.15% and Spe = 96.01% to VF, and Sen = 96.01% and Spe = 96.01% and Spe = 98.15% to VT) using Lempel-Ziv and Empirical Mode Decomposition (EMD). In this case, a selection of clean episodes of VT and VF was made too. Finally, the authors of the present work achieved high values of classification performance [47] feeding the complete time-frequency image as the input of different classifiers (e.g., Sen = 92.8% and Spe = 97.0% to VF and Sen = 91.8% and Spe = 98.7% to VF, using an Artificial Neural Network Classifier, ANNC).

In any case, the results of the TDA method in this work achieve the best results when compared with the rest of the works in the bibliography aiming to discriminate between VF and VT rhythms (despite the preselection of ECG episodes done by some works).

Table 10 shows a comparison focused on detecting VT/VF episodes, i.e., shockable and non-shockable. This set of works usually targets its implementation on external defibrillators (AED) and implantable cardioverter defibrillators (ICD). Thus, these works distinguish between shockable and non-shockable rhythms (considering shockable both VT and VF). In this group, Li et al. [71] achieved an Accuracy of Acc = 98.1% (Sen = 98.4% and Spe = 98.0%) using a Genetic Algorithm (GA) for feature selection and a SVM classifier. The same year, Alonso-Atienza et al. [70] also achieved high classification performance values (Acc = 98.6, Sen = 95.0%, and Spe = 99.0%) using a selection of features and a Support Vector Machine (SVM) classifier. This work obtained one of the highest accuracy and specificity

values in this group. In 2016, Tripathy et al. [9] used the Variational Mode Decomposition (VMD) and the Random Forest (RF) classifier to detect and classify shockable and non-shockable ECG episodes, achieving high values of accuracy, sensitivity, and specificity of 97.23%, 96.54%, and 97.97%, respectively. Later, in 2018, Mohanty et al. [11] detected and classified ventricular arrhythmias using cubic support vector machine (SVM) and C4.5 classifiers and achieving an Accuracy of Acc = 97.02% (Sen = 90.97% and Spe = 97.86%). Acharya et al. [77] brought forward an eleven-layer Convolutional Neural Network (CNN) for the classification of shockable and non-shockable arrhythmias. They obtained a 93.18% accuracy (Sen = 91.04% and Spe = 95.32%). Finally, Mohanty et al. [11] detected and classified ventricular arrhythmia using cubic support vector machine (SVM) and C4.5 classifiers, achieving high accuracy of Acc = 97.02% (Sen = 90.97% and Spe = 97.86%).

As it can be seen, the results of the TDA proposal in this work show the highest performance values also in this group of works, achieving an accuracy of 99.51%, 99.03% sensitivity, and 99.67% specificity.

Thus, the benefits of using the geometric features extracted from Topological Data Analysis (TDA) in the classification procedure are clear. Then, we can state that TDA, and the geometric features derived from it, can be successfully used both in the detection and classification of ventricular arrhythmias and in the classification of shockable episodes. It proves that the geometric features derived from Topological Data Analysis provides a good description of the signal. Moreover, it also foresees a successful application of these features in both Automated External Defibrillation (AED) and Implantable Cardioverter Defibrillation (ICD) therapies.

**Table 10.** Performance results comparison with other works discriminating shockable and non-shockable rhythms.

| Types | Shockable (VT+VF) | | | Data Base |
|---|---|---|---|---|
| Method: | Sens% | Spe% | Accu% | |
| This work, TDA | 99.03 | 99.67 | 99.51 | *AHA, MITBIH* |
| This work, PDI | 89.63 | 96.96 | 95.12 | *AHA, MITBIH* |
| [77] Convolutional neural network (CNN) (2018) | 91.04 | 95.32 | 93.18 | *CUDB, MITBIH* |
| [11] C4.5 classifier (2018) | 90.97 | 97.86 | 97.02 | *CUDB, MITBIH* |
| [78] Adaptive variational and boosted CART (2018) | 97.32 | 98.95 | 98.29 | *CUDB, MITBIH* |
| [71] SVM and bootstrap (2013) | 98.40 | 98.00 | 98.10 | *AHA, CUDB, MITBIH* |
| [9] VMD with Random Forest (2016) | 96.54 | 97.97 | 97.23 | *CUDB, MITBIH* |
| [70] FS and SVM (2013) | 95.00 | 99.00 | 98.60 | *CUDB, MITBIH* |

AHA: American Heart Association ECG Database (200 series); MIBIH: MIT-BIH Malignant Ventricular Arrhythmia Database; CCU: Registers from Coronary Care Unit (CCU) of the Royal Infirmary of Edinburgh; CUDB: MIT 'cudb' (Creighton University Ventricular Tachyarrhythmia Database).

It should be taken into account that these good results occur even in the absence of preselected ECG episodes. This work performs data classification in the same form as an Automated External Defibrillator (AED) operating in an emergency situation, following the AHA recommendations for Automated External Defibrillator (AED) algorithm performance [79]. That is, data can be continuously analyzed in time windows as they are received from the electrocardiograph.

To conclude, the success of using the TDA-derived geometric features suggests that this method may overcome conventional feature selection limitations by better describing the 'shape of data' and, thus, enabling us to build better performance arrhythmia detectors.

**Table 11.** Performance results comparison with other rhythm-discriminating works.

| Types | VF | | | VT | | | Other | | | Normal | | | Data Base |
|---|---|---|---|---|---|---|---|---|---|---|---|---|---|
| Method: | Sens% | Spe% | Accu% | Sens% | Spe% | Accu% | Sens% | Spe% | Accu% | Sens% | Spe% | Accu% | |
| This work, TDA | 97.07 | 99.25 | 98.68 | 92.72 | 99.53 | 99.05 | 97.43 | 99.54 | 99.09 | 99.05 | 98.45 | 98.76 | *AHA, MITBIH* |
| This work, PDI | 84.34 | 96.77 | 94.26 | 82.25 | 98.53 | 97.38 | 92.86 | 97.15 | 96.19 | 93.09 | 92.14 | 92.65 | *AHA, MITBIH* |
| [47] time-frequency, L2-RLR (2017) | 89.60 | 96.70 | | 91.00 | 98.10 | | 92.50 | 98.10 | | 94.90 | 96.40 | | *AHA, MITBIH* |
| [47] time-frequency, ANNC (2017) | 92.80 | 97.00 | | 91.80 | 98.70 | | 92.90 | 99.00 | | 96.20 | 96.70 | | *AHA, MITBIH* |
| [47] time-frequency, SSVR (2017) | 91.00 | 97.00 | | 92.80 | 98.70 | | 92.30 | 99.20 | | 96.60 | 96.30 | | *AHA, MITBIH* |
| [47] time-frequency, BAGG (2017) | 95.20 | 96.40 | | 88.80 | 99.70 | | 88.60 | 99.80 | | 96.60 | 94.10 | | *AHA, MITBIH* |
| [75] EMD and App Entropy (2013) | 90.47 | 91.66 | 91.17 | 90.62 | 91.11 | 90.80 | | | | | | | *MITBIH* |
| [69] TCSC algorithm (2011) | 80.97 | 98.51 | 98.14 | | | | | | | | | | *CUDB, MITBIH* |
| [76] Lempel-Ziv and EMD (2014) | 98.15 | 96.01 | | 96.01 | 98.15 | | | | | | | | *CUDB, MITBIH* |
| [68] Chaotic based (2010) | | | 88.60 | | | | | | | | | | *CCU, MITBIH* |
| [73] EMD and SVM (2019) | 99.99 | 98.40 | 99.19 | | | | | | | | | | *CUDB, MITBIH* |
| [72] CNN neural network (2017) | 56.44 | 98.19 | 97.88 | | | | | | | | | | *CUDB, MITBIH* |
| [70] FS and SVM (2013) | 91.90 | 97.10 | 96.80 | | | | | | | | | | *CUDB, MITBIH* |
| [71] Genetic algorithm, SVM (2014) | 98.40 | 98.00 | 96.30 | | | | | | | | | | *AHA, CUDB* |
| [74] Approximated entropy (2011) | 97.98 | 97.03 | | 97.03 | 97.98 | | | | | | | | *CUDB, MITBIH* |

AHA: American Heart Association ECG Database (200 series); MIBIH: MIT-BIH Malignant Ventricular Arrhythmia Database; CUDB: MIT 'cudb' (Creighton University Ventricular Tachyarrhythmia Database).

## 6. Conclusions

The rapid and reliable detection of VT and VF is fundamental in patient monitoring, but also in Automated External Defibrillation (AED) or Implantable Cardioverter Defibrillation (ICD) therapies. Any incorrect interpretation of a ventricular arrhythmia, or even the confusion between VF and VT, can be dangerous for the life of the patient.

In this paper, we propose a feature extraction method based on Topological Data Analysis (TDA) that provides near 99% accuracy in the discrimination of ventricular arrhythmias, normal and other rhythms (98.68% to VF; 99.05% to VT; 99.09% to Other; and 98.76% to Normal episodes). It also provides very high accuracy of 99.5% when discriminating between shockable (VT/VF) and non-shockable rhythms.

The novelty of this work is the incorporation of geometric features proceeding from Topological Data Analysis to the detection and classification of ventricular arrhythmias. Note also that these powerful results were obtained without preselection of episodes. Taking into consideration the obtained results, we can conclude that TDA, and the geometric features derived from it, can be successfully used both in the detection and classification of ventricular arrhythmias and in the classification of shockable rhythms. Moreover, it proves that the geometric features derived from Topological Data Analysis (TDA) provide valuable features easing the task of the classifier. Finally, we can conclude that TDA features can be beneficial in other classification tasks.

**Author Contributions:** Individual contributions: conceptualization, A.M., and A.R.-M.; methodology, A.M., A.R.-M., and M.B.-M.; software, A.M.; formal analysis, A.M., A.R.-M., and J.V.F.-V.; writing—original draft preparation, A.M., and J.V.F.-V.; writing—review and editing, A.M., J.V.F.-V., A.R.-M., and M.B.-M. All authors have read and agreed to the published version of the manuscript.

**Funding:** This research received no external funding.

**Informed Consent Statement:** Patient consent was waived due to the use of publicly available Physionet databse.

**Data Availability Statement:** Data can be publicly obtained from MIT and AHA databases, described in the references section.

**Conflicts of Interest:** The authors declare no conflict of interest.

## Abbreviations

The following abbreviations are used in this manuscript:

| | |
|---|---|
| TDA | Topological Data Analysis |
| PH | Persistent Homology |
| VF | Ventricular Fibrillation |
| VT | Ventricular Tachycardia |
| AED | Automated External Defibrillator |
| ICD | Implantable Cardioverter Defibrillator |
| ECG | Electrocardiogram |
| RF | Random Forest |
| kNN | k-Nearest Neighbor |
| ICA | Independent Component Analysis |
| DT | Decision Tree |
| PC | Point Cloud |
| RPC | Representation of Point Cloud |
| PD | Persistence Diagram |
| PDI | Persistence Diagram Image |
| KDE | Kernel Density Estimation |
| PI | Persistence Images |
| WRM | Window Reference Mark |
| TR | Time Representation |
| PDR | Persistence Diagram Representation |
| SFS | Sequential Forward Selection |

| AHA | American Heart Association |
| CUDB | Creighton University Ventricular Tachyarrhythmia Database |
| CCU | Coronary Care Unit |
| TCI | Threshold Crossing Interval |
| CNN | Convolutional Neural Network |
| EMD | Empirical Mode Decomposition |
| VMD | Variational mode decomposition |
| SVM | Support Vector Machine |
| ANNC | Artificial Neural Network Classifier |
| GA | Genetic Algorithm |

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
