# Peer review of "Ventricular Fibrillation and Tachycardia Detection Using Features Derived from Topological Data Analysis"

_applsci, doi:10.3390/app12147248_

Round 1

Reviewer 1 Report

Here are the main comments I have about the work:

  1. Grammatical errors: One very common grammatical mistake is the use of acronyms and capitalizations. Acronyms for the same work have been defined in several places (eg. TDA) but not been used later making use of the full definitions. There are several errors where capitalizations have been provided for terminology but no acronyms are done. Other smaller grammatical mistakes exist at line 8, line 9, line 28, line 61/62, line 67/68, line 182, line 380 among others
  2.  The iteration is done 5 times. The 5 is chosen as this gives the best results. Doing so leads to an optimistic bias in the output results. A number chosen independently of output should be used. I recommend 25 as this will also allow for statistical comparison of the output results from the different feature sets.
  3. Feature selection: It needs to be stated on which step the feature selection has been done. Was it done on the full data set? Or, was it done 5 times for each of the iterations? Doing on the full dataset would once again bias the results.
  4. It is mentioned 27 features were selected finally, what criteria has been used for feature selection? Is it performance on subset of training data or on the test data itself (this would again lead to bias)
  5. The comparison has been made with another feature set utilizing TDA. A benchmark feature set performance should also be done. This could be commonly used HRV or morphological feature sets.
  6. As mentioned in point 2 a significance testing needs to be performed between performance of different feature sets to strengthen the claim of improved performance. 

Please refer to the comments in the attached PDF for more details.

Author Response

Comment 1: Grammatical errors: One very common grammatical mistake is the use of acronyms and capitalizations. Acronyms for the same work have been defined in several places (eg. TDA) but not been used later making use of the full definitions. There are several errors where capitalizations have been provided for terminology but no acronyms are done. Other smaller grammatical mistakes exist at line 8, line 9, line 28, line 61/62, line 67/68, line 182, line 380 among others

Thanks for indicating the typos you found in the manuscript. They have been fixed, along with others also found in a detailed review.

Comment 2: The iteration is done 5 times. The 5 is chosen as this gives the best results. Doing so leads to an optimistic bias in the output results. A number chosen independently of output should be used. I recommend 25 as this will also allow for statistical comparison of the output results from the different feature sets.

We understand that your observation points to the acquisition of a minimum sample size to be able to use parametric tests or even statistical methods. We agree with you in this sense, however these 5 iterations actually aim to test how the model will generalize an independent dataset (it is, the ability of the model to predict new data that was not used to build the model). Thus, these 5 iterations refers to the cross-validation technique used.

One of the most used scheme for cross-validation is k-fold cross-validation. In fact, most of works of arrhythmia detection (from ECG signals) in the bibliography use 5-fold or 10-fold cross-validation. Note, as an example, that Baygin et al. [1] summarize the cross-validation techniques used in 14 relevant works of arrhythmia detection from ECG signals. As it can be seen in that work, most of them use 5 or 10-fold cross-validation.

Nevertheless, we prefer to use a repeated random sub-sampling validation because this method enables us to make the proportion of the training and validation split not dependent of the number of iterations. And the number of 5 iterations was selected because it provided the lowest generalization error.

Thus, to improve the clarity of the explanation we have rewritten and extended the description of the used cross-validation technique.

Thanks for your observation.

[1] Baygin, M.; Tucer, T.; Dogan, S.; Tan R. S.; Acharya, U.R.. Automated arrhythmia detection with fomeomorphically irreductible tree technique using more than 10,000 individual subject ECG records. Information Sciences. 2021 Oct 1. 575: 323 - 37

Comment 3: Feature selection: It needs to be stated on which step the feature selection has been done. Was it done on the full data set? Or, was it done 5 times for each of the iterations? Doing on the full dataset would once again bias the results.

Comment 4: It is mentioned 27 features were selected finally, what criteria has been used for feature selection? Is it performance on subset of training data or on the test data itself (this would again lead to bias).

Comment 5: As mentioned in point 2 a significance testing needs to be performed between performance of different feature sets to strengthen the claim of improved performance.

The feature selection was done one time, using the SFS (Sequential Forward Selection) method. This is an iterative method that begins having no features in the model and add iteratively the feature that best improves the model, until the addition of new features does not improve the performance of the model.

Following your observation, we have detected that the description corresponding to the feature selection needed to improve for the sake of clarity. According to it, the corresponding paragraph has been rewritten.

Thanks for your observation.

Comment 6: The comparison has been made with another feature set utilizing TDA. A benchmark feature set performance should also be done. This could be commonly used HRV or morphological feature sets.

This is one of the contributions of this work. For the moment there are no other works in the bibliography using topological features in arrhythmia detection. It is our proposal, so it can't be stablished a comparison with other works using topological features.

However, it has been done a comparison with other relevant studies using diverse feature sets. Tables 9 and 10 summarizes the comparison.

Comment 7: Is it performance on subset of training data or on the test data itself (this would again lead to bias).

All performance measures refer to testing performance, as the first paragraph of the 'Results' section stablish. Nevertheless, this phrase has been rewritten for the sake of clarity.

Thanks for your observation.

Thanks for your appreciations, the time and the effort dedicated to reviewing the manuscript. For sure it has served much to its improvement.

Reviewer 2 Report

The authors propose a topological data analysis to detect ventricular arrhythmias and Tachycardias in this paper.

The paper should be improved as follow:

  • Increase the resolution of the figures i.e., in Table 1 and 2 (change caption in figure 1, please), the axes are indecipherable
  • prepare a table of abbreviations used, and reduce their use where not necessary

Author Response

Comment 1: Increase the resolution of the figures i.e., in Table 1 and 2 (change caption in figure 1, please), the axes are indecipherable.

As you point out, the figures in Table 1 and 2 has been improved to provide a better visualization.

Thanks for your observation.

Comment 2: Prepare a table of abbreviations used, and reduce their use where not necessary.

The table has been included. Thanks for your observation. 

Thanks for your appreciations, the time and the effort dedicated to review the manuscript. For sure it has served to its improvement.
